# Enhancement of the Electrochemical Performances of Composite Solid-State Electrolytes by Doping with Graphene

**DOI:** 10.3390/nano12183216

**Published:** 2022-09-16

**Authors:** Xinghua Liang, Dongxue Huang, Linxiao Lan, Guanhua Yang, Jianling Huang

**Affiliations:** Guangxi Key Laboratory of Automobile Components and Vehicle Technology, Guangxi University of Science and Technology, Liuzhou 545006, China

**Keywords:** lithium-ion batteries, composite solid electrolyte, graphene, electrochemical performance

## Abstract

With high safety and good flexibility, polymer-based composite solid electrolytes are considered to be promising electrolytes and are widely investigated in solid lithium batteries. However, the low conductivity and high interfacial impedance of polymer-based solid electrolytes hinder their industrial applications. Herein, a composite solid-state electrolyte containing graphene (PVDF-LATP-LiClO4-Graphene) with structurally stable and good electrochemical performance is explored and enables excellent electrochemical properties for lithium-ion batteries. The ionic conductivity of the composite electrolyte membrane containing 5 wt% graphene reaches 2.00 × 10^−3^ S cm^−1^ at 25 °C, which is higher than that of the composite electrolyte membrane without graphene (2.67 × 10^−4^ S cm^−1^). The electrochemical window of the composite electrolyte membrane containing 5 wt% graphene reaches 4.6 V, and its Li^+^ transference numbers reach 0.84. Assembling this electrolyte into the battery, the LFP/PVDF-LATP-LiClO4-Graphene /Li battery has a specific discharge capacity of 107 mAh g^−1^ at 0.2 C, and the capacity retention rate was 91.58% after 100 cycles, higher than that of the LiFePO_4_/PVDF-LATP-LiClO_4_/Li (LFP/PLL/Li) battery, being 94 mAh g^−1^ and 89.36%, respectively. This work provides a feasible solution for the potential application of composite solid electrolytes.

## 1. Introduction

Traditional liquid lithium-ion batteries (LIBs) have potential explosion hazards due to their flammability and explosion, which hinders their commercial application in the field of energy storage and electric vehicles [1]. Solid-state electrolytes (SSEs) are highly flame-retardant; the utilization of SSEs would lead to a huge jump in the safety of LIBs. In addition, SSEs facilitate the realization of ultrathin thickness, which is expected to improve energy density and be applied in flexible and wearable devices [2,3,4]. Thus, solid-state lithium-ion batteries (SSLIBs) are considered to be the most promising next-generation secondary batteries and will replace traditional liquid LIBs.

Owing to low-cost, light-weight, environmental friendliness and high flexibility, polymer-based SSEs have been intensively investigated. Poly (vinylidene fluoride) (PVDF) is a polymer matrix with a high dielectric constant (8.4), which is beneficial to accommodate more lithium ions in the composite electrolyte and has good electrochemical stability [5]. However, single-polymer SSEs usually exhibit low electrical conductivity and poor electrochemical performance [6]. In contrast, organic–inorganic composite SSEs are highly valued in the development of solid-state batteries due to the synergistic effect of the interfacial compatibility between organic polymer SSEs and electrodes and the mechanical stability of inorganic SSEs [7,8]. NASICON-type ceramics Li_1+x_Al_x_Ti_2−x_(PO_4_)_3_ (LATP) have attracted much attention due to their high lithium-ion conductivity and air stability at room temperature and relatively low cost [9,10]. Integrating the nanoscale highly conductive inorganic particle filler LATP into the polymer electrolyte PVDF can not only reduce the crystallinity of the polymer but also increase the amorphous region and increase the motion range of the polymer chain, thereby enhancing the lithium-ion transference and improving the ionic conductivity [11] and, at the same time, maintain high stability at room temperature [12]. LiClO_4_ has good electrical conductivity and electrochemical stability. Li^+^ and ClO_4_^−^ in LiClO_4_ can interrupt the stacking of the main chain of the polymer and promote the mobility of the chain segments, thereby reducing the crystallinity of the polymer and improving the ionic conductivity [13,14]. Graphene is a two-dimensional (2D) flake composed of sp^2^-hybridized carbon atoms arranged in a honeycomb structure with special properties, and it is regarded as a component of all other graphitic carbon allotropes of different dimensions [15,16,17]. In recent years, it has been widely used in the field of lithium metal batteries [18,19]. However, the application of graphene in gel polymer electrolyte preparation has rarely been reported, and it is worth investigating the effect of graphene gel polymer electrolyte, and it is generally thought that the graphene might lead to a short circuit for its excellent electrical conductivity [20].

This work successfully prepared uniformly dispersed graphene solutions. Aiming to explore the positive effect of graphene on composite solid electrolyte membranes, we added graphene solutions of different concentrations into PVDF-LATP-LiClO_4_ (PLL) composite solid electrolyte membranes. It was found that the composite solid-state electrolyte PVDF-LATP-LiClO_4_-Graphene (PLLG) has higher lithium-ion conductivity, Li^+^ transference numbers and electrochemical window than PLL. Furthermore, the LiFePO_4_/PLLG/Li (LFP/PLLG/Li) battery has higher charge–discharge capacity, better rate capability and cycle performance than the LiFePO_4_/PLL/Li (LFP/PLL/Li) battery. This work provides an effective strategy for the application of composite solid electrolyte membranes in solid lithium batteries.

## 2. Materials and Methods

### 2.1. Preparation of Graphene

Graphene was prepared by a modified Hummers method [21]. The expanded graphite obtained by the rapid high-temperature treatment at 900 °C needs to be kept dry. An appropriate amount of concentrated H_2_SO_4_ (Aladdin, Shanghai, China) solution was dropwise introduced into a three-neck flask containing a certain mass of expanded graphite under stirring, then slowly adding concentrated HNO_3_ (Macklin, Shanghai, China), the temperature was set at 60 °C. After 10 h, a certain amount of saturated KMnO_4_ (Aladdin, Shanghai, China) solution was dropwise introduced, and the water bath was cooled to room temperature naturally after stirring for 6 h. After 6 h, an orange solution was obtained, then the graphene oxide (GO) was obtained through filtration and washing by using high-speed centrifugation. Finally, graphene oxide was reduced to obtain the graphene material required for the experiment.

### 2.2. Preparation of LATP Powder

LATP compound, one of the fast ion conductors used as SSEs, was prepared by a facile sol–gel method [22]. Lithium carbonate (Li_2_CO_3_, 99.5%, Macklin, Shanghai, China), aluminum oxide (Al_2_O_3_, 99.5%, Macklin, Shanghai, China), titanium dioxide (TiO_2_, 99%, Macklin, Shanghai, China) and diphosphate ammonium hydrogen (NH_3_H_2_PO_4_, ≥99.99%, Aladdin, Shanghai, China) were used as the precursor, weighed according to the stoichiometric ratio of Li_1.3_Al_0.3_Ti_1.7_(PO_4_)_3_. The mixture was ball milled for 8 h at the speed of 280 rpm, using ethanol as the dispersant. The obtained sample was desiccated under 80 °C for 10 h, then ground into powder of 160-mesh size. Finally, the white LATP precursor powder was obtained by annealing and calcined at 950 °C in an atmosphere furnace for 4 h.

### 2.3. Preparation of PLLG Composite Solid Electrolyte Membrane

Graphene, PVDF (Mw = 600,000, Aladdin, Shanghai, China), LATP and LiClO_4_ (≥99.5%, Aladdin, Shanghai, China) were vacuum-dried at 60 °C for 24 h. Graphene powder was dissolved in dimethylformamide (DMF, ≥99.9%, Macklin, Shanghai, China) to prepare a 5 g L^−1^ graphene–DMF solution, stirred at 45 °C for 12 h, then ultrasonically treated for 15 min. A total of 5 g of PVDF powder was dissolved in a 40 mL DMF solution and stirred at 45 °C for 1 h to form a transparent viscous solution. LiClO_4_ and LATP were introduced and stirred for 6 h. The mass ratio of PVDF, LATP and LiClO_4_ is 80:8:1. The graphene solutions of different concentrations from 0 wt% to 10 wt% were added and stirred for 12 h. Finally, the PLLG mixed solution was poured into a polytetrafluoroethylene mold and vacuum-dried at 50 °C for 24 h to obtain a PLLG composite solid electrolyte membrane. The membrane was cut into small discs with a diameter of 18 mm and stored in a glove box (argon, O_2_ < 0.1 ppm, H_2_O < 0.1 ppm) for the experiment. The preparation process is shown in Figure 1.

### 2.4. Battery Assembly

The CR2025 coin cell was assembled using LiFePO_4_ (≥99.5%, Maclean) cathode, Li anode and PLLG composite solid electrolyte. The LiFePO_4_ cathode was prepared by dissolving 80% LiFePO_4_ (≥99.5%, Macklin), 10% PVDF and 10% C (≥99.5%, Aladdin, Shanghai, China) in N-methyl pyrrolidone (NMP, AR, ≥99.5%, Macklin, Shanghai, China) solvent. After stirring for 6 h, the slurry was evenly poured onto aluminum foil. Subsequently, the aluminum foil was vacuum-dried at 60 °C for 18 h then cut into circular pole pieces with a diameter of 16 mm. The active material mass of the LiFePO_4_ positive electrode is 2.5–3 mg.

### 2.5. Physical Characterizations

The crystal structure of the prepared samples was measured on a benchtop X-ray diffractometer (XRD, D8 Advance, Bruker, Frankfurt, Germany, Cu-K, 40 kV × 30 mA). The surface structure of the materials was observed by field emission scanning electron microscopy (SEM, JSM-7001F, Osaka, Japan). Thermogravimetric analysis (TGA) was conducted using a TG analysis system (NetzschF3Tarsus, Bayern, Germany) from 30 to 800 °C with a heating rate of 10 °C min^−1^. Raman spectra were collected by a Raman microscope (ATR8000, Fujian, China). Fourier transform infrared spectra (FTIR) was performed on a Bruker Tensor 27 spectrometer (Saarbrucken, Germany). X-ray photoelectron spectroscopy (XPS) was carried out on a thermo Scientific K-Alphaxps spectrometer (New York, NY, USA).

### 2.6. Electrochemical Measurements

The impedance of the composite solid electrolyte membranes was measured using an electrochemical workstation (DH7000, Donghua, Jiangsu, China) with a stainless steel sheet (SS) as symmetric cell. The frequency range of the impedance test is 10^−1^–10^6^ Hz. The formula for calculating the ionic conductivity (*σ*) is as follows:(1)$$\sigma =\frac{L}{RS}$$
where *σ* is the ionic conductivity, L is the thickness of the composite solid electrolyte, *R* is the resistance and *S* is the contact area between the electrolyte and the test electrode (SS).

The electrochemical stability windows of the PLL and PLLG electrolyte membranes were tested using linear scanning voltammetry (LSV) by an electrochemical workstation DH7000, a lithium sheet as counter electrode and reference electrode and a stainless steel sheet as working electrode (SS). The potential range was from 2.6 V to 6 V, and the scanning rate was 0.005 V s^−1^.

The formula for the lithium-ion transference number (t*_Li_*_+_) is as follows:(2)$$t_{Li^{+}}=\frac{I_{s}\left(\Delta V-I_{0}R_{0}\right)}{I_{0}\left(\Delta V-I_{s}R_{s}\right)}$$
where Δ*V* is the applied polarization voltage. *I*_0_ and *I*_s_ are the initial value of the current and the steady-state current, respectively. *R*_0_ and *R*_s_ are the resistance initial value and steady-state resistance, respectively. For lithium symmetric cells (Li/PLL/Li and Li/PLLG/Li), the impedance was measured by AC impedance from 10^−1^ to 10^6^ Hz with an amplitude voltage of 10 mV, and the currents were measured using potentiostat amperometry with a step voltage of 50 mV.

The charge–discharge, cycle life and rate performance of the cells were performed in the voltage range of 2.8–4.2 V using a Neware tester at room temperature. EIS of the cells was performed from 10^−1^ to 10^5^ Hz using an electrochemical workstation DH7000 tester.

## 3. Results and Discussion

Figure 1 shows the chemical structures of PVDF, LiClO_4_, LATP, graphene and the preparation process of the composite solid electrolyte membrane.

Ionic conductivity is a key parameter to evaluate the mobility of lithium ions in composite solid electrolyte membranes. Figure 2a shows the EIS data of PLLG electrolytes with different contents of graphene. Each EIS spectrum consists of a semicircle at high frequency and a slopped line at low frequency. The high-frequency semicircle represents the conductivity of the bulk and intra-crystalline in the SSEs. The slopped line in low frequency is associated with lithium-ion diffusion [23]. The ionic conductivity was evaluated by Equation (1), where *R* is directly read in the real part of the *X*-axis in the Nyquist diagram and is also located at the junction of the semicircle and the sloping Li-ion diffusion line. L is the thickness of the composite solid electrolyte membrane and *S* is the effective electrode area. The R value of the PLLG electrolytes with 0 wt%, 0.5 wt%, 5 wt% and 10 wt% graphene additions are 147 Ω, 330 Ω, 74 Ω and 121 Ω, respectively. The corresponding calculated electrical conductivities are 2.67 × 10^−4^ S cm^−1^, 3.39 × 10^−4^ S cm^−1^, 2.00 × 10^−3^ S cm^−1^ and 1.32 × 10^−4^ S cm^−1^, respectively, which indicates that when the amount of graphene is 5 wt%, the ionic conductivity of the PLLG composite solid electrolyte membrane is the highest. The nonmonotonic behavior of conductivity is understandable, because too many graphene nanowires filled in the polymer matrix will lead to the aggregation and free volume depletion of nanowires, resulting in the decrease in ionic conductivity [24]. Therefore, the PLLG composite solid-state electrolyte with 5 wt% graphene was selected to compare with the PLL electrolyte to explore the potential role of graphene in the composite solid-state electrolyte membrane.

Figure 2b shows the XRD patterns of PVDF, LATP, PLL, graphene and PLLG, respectively. The characteristic diffraction peaks of graphene can be observed around 2θ = 26°, indicating that it is graphene [25]. After adding graphene, the peak intensity of PLLG was lower than that of PLL, indicating that the addition of graphene can effectively reduce the crystallinity of PLL and amorphization. In general, the amorphous region of the PVDF–LATP matrix contributes to the fixation of anions, increases the concentration of free lithium ions, promotes the segmented movement of PVDF–LATP and finally improves the ionic conductivity of solid electrolytes [26]. Figure 2c,d shows the Li^+^ transference numbers (t*_Li_*_+_) of PLL and PLLG, respectively. Li^+^ transference number (t*_Li_*_+_) is an important index to reflect the transference ability and inhibition degree of lithium dendrite cations. According to the space charge theory, lithium ions will form a concentration gradient between the negative electrode and the electrolyte, resulting in an uneven distribution of lithium ions and the formation of lithium dendrites. Therefore, the smaller t*_Li_*_+_ is, the easier it is to form strong space charge regions and serious lithium dendrites. When t*_Li_*_+_ increases, the formation of space charge and lithium dendrites can be well inhibited [27]. The present work uses the symmetrical batteries Li/PLL/Li and Li/PLLG/Li to measure t*_Li_*_+_ value. As shown in Figure 2c,d, the illustrations show the AC impedance spectra before and after measurement by chronoamperometry. The calculated Li^+^ transference numbers of PLL and PLLG are 0.58 and 0.84, respectively. As shown in Table 1, it is found that graphene promotes the Li^+^ transference numbers in the PLLG composite solid electrolyte membrane. The increase in Li^+^ transference number is attributed to the strong electrostatic interaction, which can significantly promote the dissociation of Li^+^ and ClO_4_^−^ ions, thus improving the ionic conductivity [28].

Figure 3 shows the surface microtopography and cross-sectional thickness of the electrolyte membranes. It can be seen that there are many holes of 3–4 µm on the surface of the PLL membrane (Figure 3a), while there are many small holes and grooves of about 1–2 µm on the surface of the PLLG membrane (Figure 3b), which increases the contact between the electrolyte membrane and the pole piece. The specific surface area is conducive to the transference of lithium ions. Figure 3c,e shows the cross-sectional SEM image of the PLL electrolyte membrane and the PLLG electrolyte membrane, respectively. The thicknesses of the PLL electrolyte membrane and PLLG electrolyte membrane are 203.3 µm and 250.1 µm, respectively. At the same time, it can be seen that the PLL electrolyte membrane is relatively loose, while the PLLG electrolyte membrane is denser. Figure 3d,f is an enlarged view of the cross-sectional SEM image of the PLL electrolyte membrane and PLLG electrolyte membrane, respectively. It can be seen that there are many PVDF fibers and LATP particles around 250 nm in Figure 3d,f. Moreover, the PLLG electrolyte membrane is denser than the PLL electrolyte membrane, which may be due to LATP being uniformly dispersed in the network constructed by PVDF fibers and graphene, which is more conducive to the transport of lithium ions and improves the conductivity of the composite solid-state electrolyte membrane, confirmed by the conductivity data measured in Figure 2a. Figure 3g is the EDS mapping images of the PLLG cross-section in Figure 3f, from left to right are C, O, Ti and F, respectively. It was found that graphene and LATP are uniformly dispersed in the prepared PLLG membrane.

Figure 4a shows the electrochemical window of PLL and PLLG. Obviously, the electrochemical stability window of PLLG is wider than that of PLL, the oxygen evolution voltage shifts from 4.4 V to 4.6 V after adding graphene. This is because the PLLG electrolyte forms a cross-linked network structure after adding graphene, as shown in Figure 3f, which makes the composite solid electrolyte have better mechanical properties and “softer” properties, making the PLLG electrolyte and lithium anode have better compatibility [29]. This result shows that the PLLG electrolyte does not undergo degradation or secondary redox reactions below 4.6 V, which enhances the applicability of this composite solid electrolyte. Figure 4b shows the Raman patterns of the PLL and PLLG composite solid electrolytes. The original PLL composite solid electrolyte has four distinct peaks at 1424 cm^−1^, 1547 cm^−1^, 2000 cm^−1^ and 2135 cm^−1^. After the addition of graphene, the peak intensities of these four distinct peaks in the PLLG composite solid electrolyte all weakened, and this intensity change may be due to the strong interaction between the dissociated salt and the mixed polymer matrix. The addition of anions from the salt to the polymer matrix helps to improve the amorphous nature of the polymer, reduces the crystallinity of the host polymer, and improves the ionic conductivity of the composite solid electrolyte membrane due to its role as a plasticizer [30]. Figure 4c shows the FTIR spectra of the PLL and PLLG composite solid electrolytes at 600–1700 cm^−1^. The absorption peaks at 779, 852, 913, 1123, 1312, 1445 and 1594 cm^−1^ belong to the typical stretching and bending modes of PVDF. Apparently, the peaks at 623 and 1670 cm^−1^ are attributed to the partial dehydrogenation of PVDF to C=C, indicating a strong interaction between graphene nanofibers and PLL [31,32]. Figure 4d shows the TG analysis of PLL and PLLG composite solid electrolytes. There are two weight loss processes on the TG curves of the PLLG composite solid electrolyte, which are about 150 and 480 °C, respectively. The first weight loss is caused by the partial oxidative decomposition of graphene, as well as water molecules in the composite solid electrolyte membrane having been removed [33], and the weight loss rate is 45.89%. The second weight loss is 33.846%, mainly due to the decomposition of PVDF, and the weight degradation of the polymer PVDF is caused by the volatilization of hydrogen fluoride and the oxidative decomposition of fluorocarbon organic compounds. For the TG curves of the PLL composite solid electrolyte membrane, only the second weight loss can be observed, and the weight loss rate is 66.43%, much higher than that of the PLLG composite solid electrolyte. The TG analysis indicates that the addition of graphene inhibits the decomposition of PVDF to a certain extent, which improves the thermal stability of the PLLG membrane.

As shown in Figure 5a, two prominent peaks representing C 1s and O 1s were observed at 285 and 532 eV on the XPS spectrum, respectively. Compared with the spectra of graphene, there are Ti3p (32 eV) and F1s (688 eV) in PLL and PLLG, indicating that PVDF and LATP are stable after graphene addition. In addition, as shown in Figure 5b,c, after adding graphene, C=O in PLLG decreases, while C-C and C-O increase, which is conducive to the improvement of ionic conductivity.

Figure 6a shows the specific charge–discharge capacity of LFP/PLL/Li and LFP/PLLG/Li at 0.1 C. The discharge voltage plateau of both LFP/PLL/Li and LFP/PLLG/Li is about 3.40 V, and the charging voltage plateau is about 3.46 V. The specific charge capacity and discharge capacity of LFP/PLLG/Li are 141 mAh g^−1^ and 133 mAh g^−1^, respectively, higher than that of LFP/PLL/Li, being 121 mAh g^−1^ and 116 mAh g^−1^, respectively. Figure 6b shows the rate capability of LFP/PLL/Li and LFP/PLLG/Li. The specific discharge capacities of LFP/PLL/Li at 0.1 C, 0.2 C, 0.5 C, 1 C and return 0.1 C are 116, 94, 67, 46 and 118 mAh g^−1^, respectively. In contrast, the rate capability of LFP/PLLG/Li is better than that of LFP/PLL/Li, and its specific discharge capacities are 133, 107, 78, 57 and 118 mAh g^−1^, respectively. Figure 6c shows the charge–discharge curves of LFP/PLL/Li at the 1st, 5th, 10th, 50th and 100th cycles at 0.2 C, and the corresponding specific capacities are 94, 87, 88, 87 and 84 mAh g^−1^, respectively, lower than that of LFP/PLLG/Li, being 108, 108, 104, 99 and 98 mAh g^−1^, respectively, as shown in Figure 6d.

The above experiment results demonstrate that the LFP/PLLG/Li full battery shows significantly better electrochemical performance than that of the LFP/PLL/Li full battery, regardless of the first charge–discharge specific capacity or rate performance.

Figure 7a shows the cycling stability and coulombic efficiency of LFP/PLL/Li and LFP/PLLG/Li at 0.2 C. It is known that the applied temperature has an obvious influence on the electrochemical performances of the batteries, so the discharge retention graphs in Figure 7a are not linear in shape and may be due to the temperature in the test room not being strictly stable. However, it should be noted that the discharge capacity of LFP/PLLG/Li is larger than LFP/PLL/Li in 100 cycles. After 100 cycles, the discharge capacity of LFP/PLLG/Li and LFP/PLL/Li are 98 and 84 mAh g^−1^, respectively, with corresponding capacity retention rates of 91.59% and 89.36%, respectively, indicating that the addition of graphene improved the cycling performance of the composite electrolyte membranes.

Figure 7b shows the EIS spectra of LFP/PLL/Li and LFP/PLLG/Li before and after 100 cycles at 0.2 C. These spectra were interpreted on the basis of the simple equivalent circuit model (shown in the inset in Figure 7b). The intercept of the *Z*’-axis at high frequency is associated with the ohmic resistance Rb, and Rct and Zw represent the charge–transfer resistance at the electrolyte/electrode interface and the Warburg impedance for the diffusion of lithium ions in electrodes, respectively. Moreover, the constant phase element (CPE) is used instead of a pure capacitive element (Cdl) [34]. The Rct values of LFP/PLL/Li before and after 100 cycles are 436 Ω and 116.5 Ω, respectively. The Rct values of LFP/PLLG/Li before and after 100 cycles are 239 Ω and 107.5 Ω, respectively. This result indicates that graphene addition enhances the conduction of lithium ions between the composite solid electrolyte membrane and the electrode.

## 4. Conclusions

In summary, we demonstrated that the addition of graphene improves the electrochemical performance of the composite solid electrolyte, thereby significantly improving the electrochemical performance of the battery. With 5 wt% graphene addition, the ionic conductivity, electrochemical window and Li^+^ transference numbers of the PLLG composite solid electrolyte membrane reach 2 × 10^−3^, 4.6 V and 0.84, respectively. Assembled into a complete battery with this composite solid electrolyte membrane, the LFP/PLLG/Li battery shows better electrochemical performance than the LFP/PLL/Li battery. The specific discharge capacity of LFP/PLLG/Li is 133 mAh g^−1^ at 0.1 C, higher than that of LFP/PLLG/Li (116 mAh g^−1^). In addition, the LFP/PLLG/Li battery shows better cycling stability than the LFP/PLL/Li battery. After 100 cycles at 0.2 C, the capacity retention rates of the LFP/PLLG/Li battery and the LFP/PLL/Li battery are 91.59% and 89.36%, respectively. These results reveal a promising future for exploring more solid electrolyte materials for solid-state batteries.

## Figures and Tables

**Figure 1 nanomaterials-12-03216-f001:**
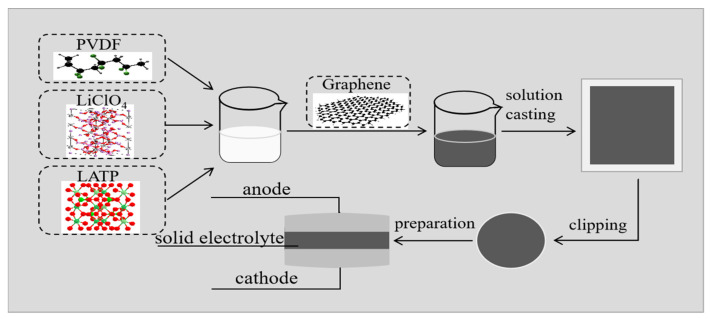
Schematic diagram of the preparation of the poly (vinylidene fluoride)-Li_1+x_Al_x_Ti_2−x_(PO_4_)_3_-LiClO_4_-Graphene (PLLG) composite solid electrolyte.

**Figure 2 nanomaterials-12-03216-f002:**
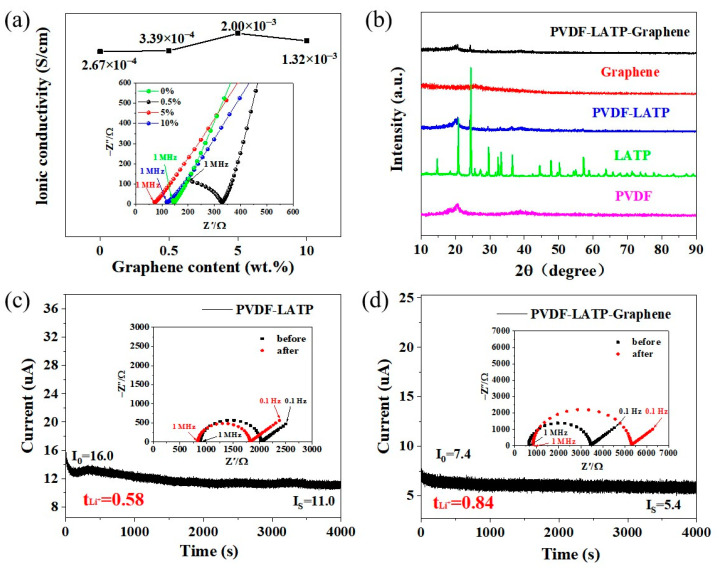
(**a**) Ionic conductivity of PLLG electrolyte membranes, (**b**) X-ray diffraction spectra of PVDF, LATP, graphene, PLL and PLLG electrolyte membranes, (**c**) Li^+^ transference number of PLL electrolyte membrane and (**d**) Li^+^ transference number of PLLG electrolyte membrane.

**Figure 3 nanomaterials-12-03216-f003:**
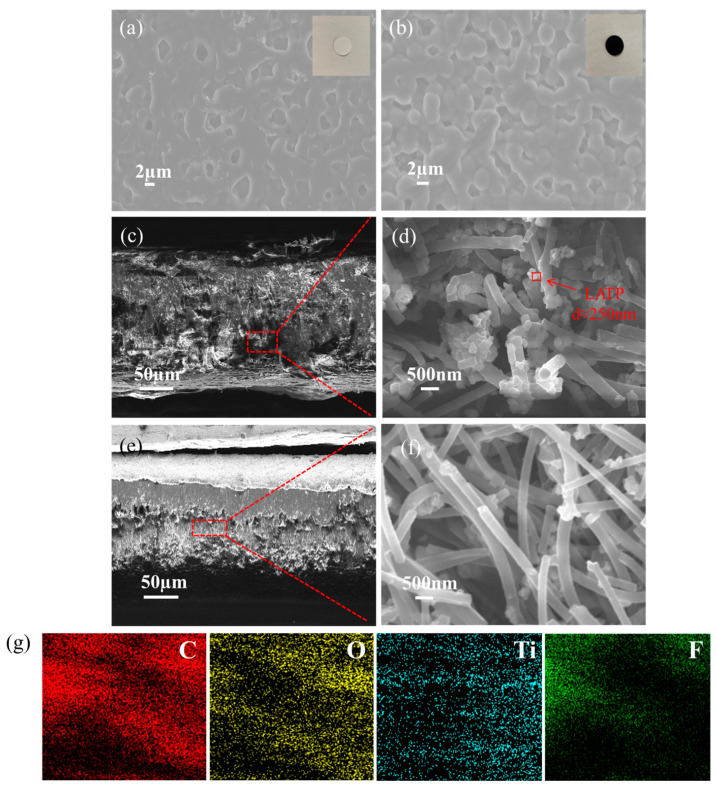
SEM image of PLL SSE (**a**) and PLLG SSE (**b**), cross-sectional SEM images of PLL SSE (**c**,**d**) and LLG SSE (**e**,**f**) and EDS mapping images of PLLG (**g**).

**Figure 4 nanomaterials-12-03216-f004:**
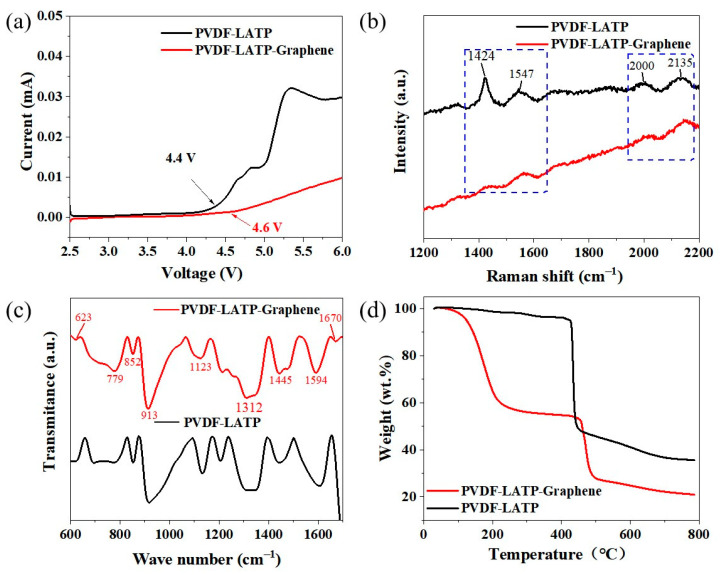
(**a**) Electrochemical windows, (**b**) Raman spectra, (**c**) FTIR spectra and (**d**) TG diagram of the PLL and PLLG composite solid electrolyte.

**Figure 5 nanomaterials-12-03216-f005:**
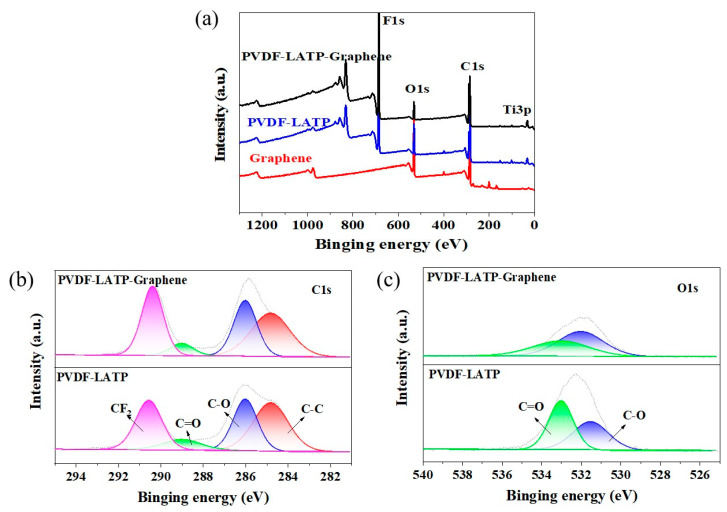
(**a**) XPS spectra of graphene, PLL and PLLG: (**b**) C 1s and (**c**) O 1s.

**Figure 6 nanomaterials-12-03216-f006:**
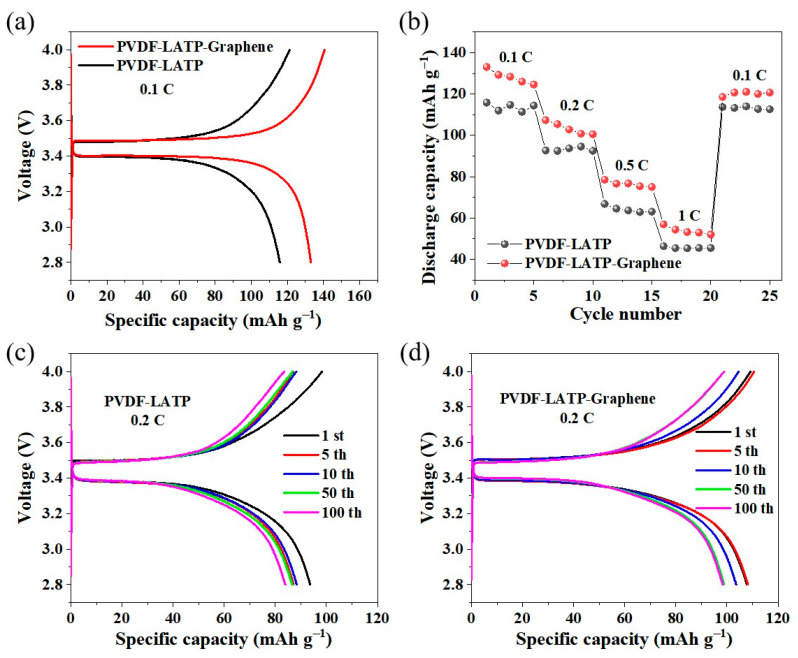
(**a**) First charge–discharge curves of LFP/PLL/Li and LFP/PLLG/Li at 0.1 C, (**b**) rate capability of LFP/PLL/Li and LFP/PLLG/Li, charge–discharge curves of (**c**) LFP/PLL/Li and (**d**) LFP/PLLG/Li at the 1st, 5th, 10th, 50th and 100th cycles at 0.2 C.

**Figure 7 nanomaterials-12-03216-f007:**
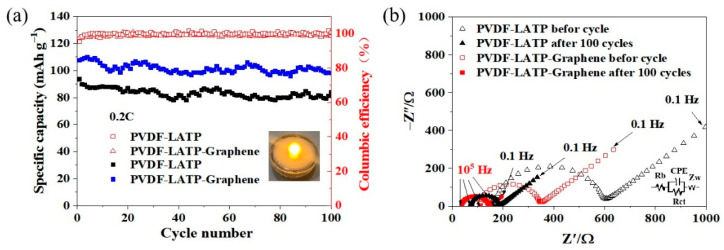
(**a**) Cycling stability and coulomb efficiency of LFP/PLL/Li and LFP/PLLG/Li at 0.2 C and (**b**) EIS diagram of LFP/PLL/Li and LFP/PLLG/Li before and after 100 cycles.

**Table 1 nanomaterials-12-03216-t001:** Li^+^ transference numbers of PLL and PLLG electrolyte membranes.

	Electrolytes	
	PLL	PLLG
I_0_/µA	16	7.4
I_s_/µA	11	5.4
R_0_/Ω	2040	3450
R_s_/Ω	1820	5300
V/mV	50	50
t_Li+_	0.58	0.84

## Data Availability

Not applicable.

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
