# Peer review of "Enhancement of the Electrochemical Performances of Composite Solid-State Electrolytes by Doping with Graphene"

_nanomaterials, 2022, doi:10.3390/nano12183216_

Round 1

Reviewer 1 Report

The overall quality of the work that is presented in this manuscript is excellent.  The scientific content will be interesting to others who are researching ways to enhance battery performance.  Here are a few minor revisions to be made:

1. Line 129: 'electrochemical impedance Spectroscopy (EIS)' should be replaced with 'resistance'.

2. Line 156 and 157: Replace 'slopy' with 'slopped'.

3. Font size changes at Line 209.

4. Font size changes at Line 273.

5. The plots in Figure 2 are too small and should be enlarged for improved readability.

6. Table 1 Shows 'Electrolyte' over the first column. However, this is a little confusing upon first look because it looks like the column header.  To improve clarity, 'Electrolyte' should be moved and centered above the columns 'PLL' and 'PLLG'.

7. In the paragraph starting with line 298, resistance is discussed.  Figure 7 (b) shows semicircles that are decreasing in diameter for the different compositions before and after cycling which indicates that internal resistance is decreasing.  The paper could be improved by answering the following question: 'Why is the battery's internal resistance decreasing after extended battery cycling?" 

8. The section, '4. Conclusions', could be made more declarative to finish-off stronger.  However, this is just a matter of style and therefore just a note to consider.

Author Response

We greatly thank the reviewer for the positive evaluation and valuable comments on the previous manuscript. In the new version, we have revised our manuscript according to the reviewer’s comments. The following attachment is our response to the reviewer.

Reviewer 2 Report

The manuscript describes “Enhancement of the electrochemical performance of composite solid-state electrolytes by doping with graphene”

The analysis of obtained data does not seem to be enough. The authors need minor revision to be accepted in the journal.

1.   Authors need to adjust figures’ resolution and to increase Font size of inserted text in Figures(Figure2, Figure4, Figure5, Figure6, and Figure7)

2.   Authors showed Figure 7(a) for discharge capacities at 0.1C and 0.2C. The discharge retention graphs are not linear shape compared to conventional lithium ion battery using liquid electrolyte.

Authors need to explain the reason of the nonlinear shape of discharge capacities

3. Authors showed Figure 7(b) for impedance plots.

The data showed that The impedance result of PVDF-LATP-Graphene after 100 cycles improved compared to that of PVDF-LATP-before cycles. In usual, as increase cycle numbers, the impedance data(resistance) become large. However, for the Figure 7(b) data, as increase cycle number, the impedance data decrease.

4.   Authors showed Figure 6 (d) for the specific capacities for 1st, 5nd, 10th, 50th, and 100th. The graph showed the similar values between 50th and 100th. The values are not coincident between figure 6(d) and figure 7(a).

Author should explain about coincident between figure 6(d) and figure 7(a).   

Author Response

(The authors gave the same response as above.)
